# BANDITS WITH RANKING FEEDBACK

## ABSTRACT

In this paper, we introduce a novel variation of multi-armed bandits called *bandits with ranking feedback*. Unlike traditional bandits, this variation provides feedback to the learner that allows them to rank the arms based on previous pulls, without quantifying numerically the difference in performance. This type of feedback is well-suited for scenarios where the arms' values cannot be precisely measured using metrics such as monetary scores, probabilities, or occurrences. Common examples include human preferences in matchmaking problems. Furthermore, its investigation answers the theoretical question on how numerical rewards are crucial in bandit settings. In particular, we study the problem of designing *no-regret* algorithms with ranking feedback both in the *stochastic* and *adversarial* settings. We show that, with stochastic rewards, differently from what happens with non-ranking feedback, no algorithm can suffer a logarithmic regret in the time horizon $T$ in the instance-dependent case. Furthermore, we provide two algorithms. The first, namely DREE, guarantees a superlogarithmic regret in $T$ in the instance-dependent case thus matching our lower bound, while the second, namely R-LPE, guarantees a regret of $\widetilde{\mathcal{O}}(\sqrt{T})$ in the instance-independent case. Remarkably, we show that no algorithm can have an optimal regret bound in both instance-dependent and instance-independent cases. We also prove that no algorithm can achieve a sublinear regret when the rewards are adversarial. Finally, we numerically evaluate our algorithms in a testbed, and we compare their performance with some baseline from the literature.

## 1 INTRODUCTION

*Multi-armed bandits* are well-known sequential decision-making problems where a learner is given a number of arms whose reward is unknown (Lattimore & Szepesvari, 2017). At every round, the learner can pull an arm and observe a realization of the reward associated with that arm, which can be generated *stochastically* (Auer et al., 2002) or *adversarially* (Auer et al., 1995). The central question in multi-armed bandits concerns how to address the *exploration/exploitation* tradeoff to minimize the *regret* between the reward provided by the *learning policy* and the optimal *clairvoyant* algorithm. Interestingly, multi-armed bandits come with several flavors capturing a wide range of different applications, *e.g.*, with delayed feedback (Vernade et al., 2017; 2020), combinatorial constraints (Combes et al., 2015), and a continuous set of arms (Kleinberg et al., 2019).

In this paper, we introduce a novel variation of multi-armed bandits that, to the best of our knowledge, is unexplored so far. We name the model as *bandits with ranking feedback*. This feedback provides the learner with a partial observation over the rewards given by the arms. More precisely, the learner can rank the arms based on the previous pulls they experienced, but they cannot quantify numerically the difference in performance. Thus, the learner is not allowed to asses how much an arm is better or worse than another. This type of feedback is well-suited for scenarios where the arms' values cannot be precisely measured using metrics such as monetary scores, probabilities, or occurrences, and naturally applies to various settings, *e.g.*, when dealing with human preferences such as in matchmaking settings among humans and when the scores cannot be revealed for privacy or security reasons. This latter case can be found, *e.g.*, in online advertising platforms offering automatic bidding services as they have no information on the actual revenue of the advertising campaigns since the advertisers prefer not to reveal these values being sensible data for the companies.[1] Remarkably, our

---

[1]Notice that a platform can observe the number of clicks received by an advertising campaign, but it cannot observe the revenue associated with that campaign.

model poses the interesting theoretical question whether the lack of numerical scores precludes the design of sublinear regret algorithms or worsens the regret bounds that are achievable when numerical scores are available.

**Related Works.** The field most related to bandits with ranking is *preference learning*, which aims at learning the preferences of one or more agents from some observations (Fürnkranz & Hüllermeier, 2010). Let us remark that preference learning has recently gained a lot of attention from the scientific community, as it enables the design of AI artifacts capable of interacting with human-in-the-loop (HTL) environments. Indeed, human feedback may be quite misleading when it is asked to report numerical values, while humans are far more effective at reporting ranking preferences. The preference learning literature mainly focuses on two kinds of preference observations: pairwise preferences and ranking. In the first case, the data observed by the learner involves preferences between two objects, *i.e.*, a partial preference is given to the learner. In the latter, a complete ranking of the available data is given as feedback. Our work belongs to the latter branch. Preference learning has been widely investigated by the online learning community, see, *e.g.*, (Bengs et al., 2018).

Precisely, our work presents several similarities with the *dueling bandits* settings (Yue et al., 2012; Saha & Gaillard, 2022; Lekang & Lamperski, 2019), where, in each round, the learner pulls two arms and observes a ranking over them. Nevertheless, although dueling bandits share similarities to our setting, they present substantial differences. Specifically, in our model, the learner observes a ranking depending on the arms they have pulled so far. In dueling bandits, the learner observes an instantaneous comparison between the arms they have just pulled; thus, the outcome of such a comparison does not depend on the arms previously selected, as is the case of bandits with ranking feedback. As a consequence, while in bandits with ranking feedback the goal of the learner is to exploit the arm with the highest mean, in dueling bandits the goal of the learner is to select the arm winning with the highest probability. Furthermore, while we adopt the classical notion of regret used in the bandit literature to assess the theoretical properties of our algorithms, in dueling bandits, the algorithms are often evaluated with a suitable notion of regret, which differs from the classical one.

Dueling bandits have their reinforcement learning (RL) counterpart in the *preference-based reinforcement learning* (PbRL), see, *e.g.*, (Novoseller et al., 2019) and (Wirth et al., 2017). Interestingly, PbRL techniques differ from the standard RL approaches in that they allow an algorithm to learn from non-numerical rewards; this is particularly useful when the environment encompasses human-like entities (Chen et al., 2022). Furthermore, PbRL provides a bundle of results, ranging from theory (Xu et al., 2020) to practice (Christiano et al., 2017; Lee et al., 2021). In PbRL, preferences may concern both states and actions; contrariwise, our framework is stateless since the rewards gained depend only on the action taken during the learning dynamic. Moreover, the differences outlined between dueling bandits and bandits with ranking feedback still hold for preference-based reinforcement learning, as preferences are considered between observations instead of the empirical mean of the accumulated rewards.

**Original Contributions.** We investigate the problem of designing *no-regret* algorithms for bandits with ranking in both *stochastic* and *adversarial* settings. With stochastic rewards, we show that ranking feedback does not preclude sublinear regret. However, it worsens the upper bounds achievable by the algorithms. In particular, in the instance-dependent case, we show that no algorithm can suffer from a logarithmic regret in the time horizon (as instead is possible in the non-ranking case), and we provide an algorithm, namely DREE (Dynamical Ranking Exploration-Exploitation), guaranteeing superlogarithmic regret that matches the lower bound. In the instance-independent case, a crucial question is whether there is an algorithm providing a regret bound better than the well-known Explore-then-Commit algorithm which trivially guarantees a regret of $\tilde{\mathcal{O}}(T^{2/3})$ in our case. We design an algorithm, namely R-LPE (Ranking Logarithmic Phased Elimination), which guarantees a regret of $\tilde{\mathcal{O}}(\sqrt{T})$ in the instance-independent case. More importantly, we show that no algorithm can have an optimal regret bound in both instance-dependent and instance-independent cases. Furthermore, with adversarial rewards, we show that ranking feedback precludes sublinear regret, and therefore numerical rewards are strictly necessary in adversarial online learning settings. Finally, we numerically evaluate our DREE and R-LPE algorithms in a testbed, and we compare their performance with some baseline from the literature in different settings. We show that our algorithms dramatically outperform the baselines in terms of empirical regret.

## 2 PROBLEM FORMULATION

In this section, we formally state the model of bandits with ranking feedback and discuss the learner-environment interaction. Subsequently, we define policies and the regret notion both in the *stochastic* and in the *adversarial* settings.

**Setting and Interaction.** Differently from standard bandits—see, *e.g.*, the work by (Lattimore & Szepesvari, 2017)—in which the learner observes the *reward* associated with the pulled arm, in bandits with ranking feedback the learner can only observe a *ranking* over the arms based on the previous pulls. Formally, we assume the learner-environment interaction to unfold as follows.[2]

(i) At every round $t \in [T]$, where $T$ is the time horizon, the learner chooses an arm $i_t \in \mathcal{A} := [n]$, where $\mathcal{A}$ is the set of available arms and $n = |\mathcal{A}| < +\infty$.

(ii) We study both stochastic and adversarial rewards. In the stochastic setting, the environment draws the reward $r_t(i_t)$ associated with arm $i_t$ from a probability distribution $\nu_{i_t}$, *i.e.*, $r_t(i_t) \sim \nu_{i_t}$, whereas, in the adversarial setting, $r_t(i_t)$ is chosen adversarially by an opponent from a bounded set of reward functions.

(iii) There is a bandit feedback on the reward of the arm $i_t \in \mathcal{A}$ pulled at round $t$ leading to the estimate of the empirical mean of $i_t$ as follows:

$$\hat{r}_t(i) := \frac{\sum_{j \in \mathcal{W}_t(i)} r_j(i)}{Z_i(t)},$$

where $\mathcal{W}_t(i) := \{\tau \in [t] \mid i_\tau = i\}$ and $Z_i(t) := |\mathcal{W}_t(i)|$.[3] However, the learner observes the rank over the empirical means $\{\hat{r}_t(i)\}_{i \in \mathcal{A}}$ We denote with $\mathcal{S}_\mathcal{A}$ the set containing all the possible permutations of the elements of set $\mathcal{A}$. Formally, we assume that the ranking $\mathcal{R}_t \in \mathcal{S}_\mathcal{A}$ observed by the learner at round $t$ is such that:

$$\hat{r}_t(\mathcal{R}_{t,i}) \geq \hat{r}_t(\mathcal{R}_{t,j}) \ \forall t \in [T] \ \forall i, j \in [n] \text{ s.t. } i \geq j,$$

where $\mathcal{R}_{t,i} \in \mathcal{A}$ denotes the $i$-th element in the ranking $\mathcal{R}_t$ at round $t \in [T]$.

For the sake of clarity, we provide an example to illustrate bandits with ranking feedback and the corresponding learner-environment interaction.

**Example.** We consider an environment with two arms, *i.e.*, $\mathcal{A} = \{1, 2\}$, in which the learner plays the first action at rounds $t = 1$ and $t = 3$ and the second action at round $t = 2$, so that $\mathcal{W}_3(1) = \{1, 3\}$ and $\mathcal{W}_3(2) = \{2\}$. Let $r_1(1) = 1$ and $r_3(1) = 5$ be the rewards when playing the first arm at rounds $t = 1$ and $t = 3$, respectively, while let $r_2(2) = 5$ be the reward when playing the second arm at round $t = 2$. The empirical means of the two arms and resulting rankings at every round $t \in [3]$ are given by:

$$\begin{cases} \hat{r}_t(1) = 1, \ \hat{r}_t(2) = 0 \ \mathcal{R}_t = \langle 1, 2 \rangle & t = 1 \\ \hat{r}_t(1) = 1, \ \hat{r}_t(2) = 5 \ \mathcal{R}_t = \langle 2, 1 \rangle & t = 2 \\ \hat{r}_t(1) = 3, \ \hat{r}_t(2) = 5 \ \mathcal{R}_t = \langle 2, 1 \rangle & t = 3 \end{cases}.$$

**Policies and Regret.** At every round $t$, the action played by the learner is prescribed by a policy $\pi$. In both the stochastic and adversarial settings, we let the policy $\pi$ be a randomized map from the history of the interaction $H_{t-1} = (\mathcal{R}_1, i_1, \mathcal{R}_2, i_2, \ldots \mathcal{R}_{t-1}, i_{t-1})$ to the set of all the probability distributions with support $\mathcal{A}$. Formally, we let $\pi : H_{t-1} \to \Delta(\mathcal{A})$, for $t \in [T]$, such that $i_t \sim \pi(H_{t-1})$. As it is customary in bandits, the learner's goal is to design a policy $\pi$ minimizing the cumulative expected regret, whose formal definition is as follows:

$$R_T(\pi) = \mathbb{E}\left[\sum_{t=1}^{T} r_t(i^*) - r_t(i_t)\right],$$

---

[2]Given $n \in \mathbb{N}_{>0}$ we denote with $[n] := \{1, \ldots, n\}$.
[3]Note that the latter definition is well-posed as long as $|\mathcal{W}_t(i)| > 0$. For each $i \in \mathcal{A}$ and $t \in [T]$ such that $|\mathcal{W}_t(i)| = 0$, we let $\hat{r}_t(i) = 0$.

where the expectation is over the randomness of both the policy and environment in the stochastic setting, and we let $i^* \in \arg\max_{i \in \mathcal{A}} \mu_i$ with $\mu_i = \mathbb{E}[\nu_i]$, whereas the expectation is over the randomness of the policy in the adversarial setting and we let $i^* \in \arg\max_{i \in \mathcal{A}} \sum_{t=1}^{T} r_t(i)$. For the sake of simplicity, from here on, we omit the dependence on $\pi$, referring to $R_T(\pi)$ as $R_T$. The impossibility of observing the reward realizations raises several technical difficulties when designing no-regret algorithms since the approaches adopted for standard (non-ranking) bandits do not generalize to our case. In the following sections, we discuss how the lack of this information degrades the performance of the algorithms when the feedback is ranking.

## 3 ANALYSIS IN THE STOCHASTIC SETTING

Initially, we observe that approaches based on *optimism-vs.-uncertainty*, such as UCB1, might be challenging to apply within our framework. This is because the learner lacks the information to estimate the reward associated with an arm, making it difficult to infer a confidence bound. Therefore, the most popular class of algorithms one can employ in bandits with ranking feedback is that of *explore-then-commit* (EC) algorithms, where the learner either exploits a single arm or explores the others according to a deterministic or randomized exploration strategy.

In the following, we distinguish the instance-dependent case from the instance-independent one. In particular, we provide two algorithms, each guaranteeing a sublinear regret in one of the two cases.

### 3.1 INSTANCE-DEPENDENT LOWER BOUND

It is well-known that standard bandits admit algorithms guaranteeing a regret that is logarithmic in time horizon $T$ in the instance-dependent case. We show in this section that such a result does not hold when the feedback is provided as a ranking. More precisely, our result rules out the possibility of having a logarithmic regret. However, in the next section, we prove that we can get a regret whose dependence on $T$ is arbitrarily close to a logarithm, thus showing that the extra cost one has to pay in the instance-dependent case to deal with ranking feedback is asymptotically negligible in $T$.

Our impossibility result exploits a connection between random walks and arms' cumulative rewards. Formally, we define an (asymmetric) random walk as follows.

**Definition 1.** *A random walk is a stochastic process $\{G_t\}_{t \in \mathbb{N}}$ such that:*

$$G_t = \begin{cases} 0 & t = 1 \\ G_{t-1} + \epsilon_t & t > 1 \end{cases},$$

*where $\{\epsilon_t\}_{t \in \mathbb{N}}$ is an i.i.d. sequence of random variables, and $\mathbb{E}[\epsilon_t]$ is the drift of the random walk.*

We model the cumulative reward collected by a specific arm during the learning process as a random walk, where the drift represents the expected reward associated with that arm. Let us notice that, in bandits where the feedback is not given as a ranking, the learner can completely observe the evolution of the random walks, being able to observe the realizations of the reward associated with each pulled arm. Such observations allow the learner to estimate the difference between the performance of each pair of arms. For instance, the learner can observe whether two arms perform similarly or, instead, whether the gap between their performances is significant. Differently, in our case, the learner only observes the rank without quantify numerically the performance.

This loss of information raises several technical issues that are crucial, especially when the random walks never switch. Intuitively, in bandits with ranking feedback, we can observe how close the expected rewards of two arms are only by observing subsequent switches of their positions in the ranking. However, there is a strictly positive probability that two random walks never switch (thus leading to no intersection) when they have a different drift $\mathbb{E}[\epsilon_t]$ and therefore we may not evaluate how two arms are close. This is shown in the following lemma.

**Lemma 1** (Separation lemma). *Let $G_t, G'_t$ be two independent random walks defined as:*

$$G_{t+1} = G_t + \epsilon_t \qquad and \qquad G'_{t+1} = G'_t + \epsilon'_t,$$

*where $G_0 = G'_0 = 0$ and the drifts satisfy $\mathbb{E}[\epsilon_t] = p > q = \mathbb{E}[\epsilon'_t]$,. Then:*

$$\mathbb{P}\left(\forall t, t' \in \mathbb{N}^* \ G_t/t \geq G'_{t'}/t'\right) > 0.$$

The rationale of the above lemma is that, given two random walks with different drifts, there is a line separating them with a strictly positive probability. Therefore, with a non-negligible probability, the empirical mean corresponding to the process with the higher drift upper bounds forever the empirical mean of the process with the lower drift. In bandits with ranking feedback, such a separation lemma shows that the problem of distinguishing two different instances is harder than in the standard, non-ranking feedback case. Before stating our result, as is customary in bandit literature, let us denote with $\Delta_i := \mu_i^* - \mu_i$, where we let $i^* \in \arg\max_{i \in \mathcal{A}} \mu_i$ and $\mu_i := \mathbb{E}[\nu_i]$. Now, we can state the following result for the instance-dependent case.

**Theorem 1** (Instance-dependent lower bound). *Let $\pi$ be any policy for the bandits with ranking feedback, then, for any $C(\cdot) : [0, +\infty) \to [0, +\infty)$, there is $\{\Delta_i\}_{i \in [n]}$ and a time horizon $T > 0$ such that $R_T > \sum_{i=1}^n C(\Delta_i) \log(T)$.*

*Proof sketch.* It is well-known in the bandit literature that, to achieve logarithmic regret, it is necessary to pull any suboptimal arm at least $\sim \frac{\log(T)}{\Delta_i^2}$ times. The values of $\Delta_i$ cannot be estimated without a switch in the ranking. Since even when $\Delta_i$s are very small, the optimal arm may remain in the first position for the whole process, $\Delta_i$ cannot be estimated, and it is necessary to pull the suboptimal arms more than $\mathcal{O}(\log(T))$ times. $\qquad\qquad\square$

### 3.2 INSTANCE-DEPENDENT UPPER BOUND

We introduce the Dynamical Ranking Exploration-Exploitation algorithm (DREE). The pseudo-code is provided in Algorithm 1. As usual in bandit algorithms, in the first $n$ rounds, a pull for each arm is performed (Lines 2–4). At every subsequent round $t > n$, the exploitation/exploration tradeoff is addressed by playing the best arm according to the received feedback unless there is at least one arm whose number of pulls at $t$ is smaller than a superlogarithmic function $f(t) : (0, \infty) \to \mathbb{R}_+$.[4] More precisely, the algorithm plays an arm $i$ at round $t$ if it has been pulled less than $f(t)$ times (Lines 5–6), where ties due to multiple arms pulled less than $f(t)$ times are broken arbitrarily. Instead, if all arms have been pulled at least $f(t)$ times, the arm in the highest position of the last ranking feedback is pulled (Lines 7–9). Each round ends once the learner receives the feedback in terms of ranking over the arms (Line 10). Let us observe that the exploration strategy of Algorithm 1 is deterministic, and the only source of randomness concerns the realization of the arms' rewards.

---

**Algorithm 1** Dynamical Ranking Exploration-Exploitation (DREE)

---

1: **for** $t \in [T]$ **do**
2:   **if** $t \leq n$ **then**
3:    play arm $i_t$
4:   **end if**
5:   **if** There is an arm $i$ played less than $f(t)$ times **then**
6:    Play $i_t = i$
7:   **else**
8:    Play $i_t = \mathcal{R}_{t-1,1}$
9:   **end if**
10:   Receive updated ranking $\mathcal{R}_t$
11: **end for**

---

We state the following result, providing the upper regret bound of Algorithm 1 as a function of $f$.

**Theorem 2** (Instance-dependent upper bound). *Assume that the reward distribution of every arm is 1-subgaussian. Let $f : (0, \infty) \to \mathbb{R}$ be a superlogarithmic function in t, then there is a term $C(f, \Delta_i)$ for each sub-optimal arm $i \in [n]$ which does not depend on $T$, such that Algorithm 1 satisfies:*

$$R_T \leq f(T) \sum_{i=1}^n \Delta_i + \log(T) \sum_{i=1}^n C(f, \Delta_i).$$

---

[4]A function $f(t)$ is superlogarithmic when $\lim_{t \to \infty} \frac{f(t)}{\log(t)} = +\infty$.

To minimize the asymptotic dependence in $T$ of the cumulative regret suffered by the algorithm, we can choose, *e.g.*, $f : (0, \infty) \to \mathbb{R}$ as $f(t) = \log(t)^{1+\delta}$, where parameter $\delta > 0$ is as small as possible. However, the minimization of $\delta$ comes at the cost of increasing the terms $C(f, \Delta_i)$ as they grow exponentially as $\delta > 0$ goes to zero as long as $\Delta_i < 1$. In particular, the terms $C(f, \Delta_i)$ are defined as stated in the following corollary.

**Corollary 3.** *Let $\delta > 0$ and $f(t) = \log(t)^{1+\delta}$ be the sperlogarithmic function used in Algorithm 1, then we have:*

$$C(f, \Delta_i) = \frac{2\Delta_i \left( e^{\left( \left( 2/\Delta_i^2 \right)^{1/\delta} \right)} + 1 \right)}{1 - e^{-\Delta_i^2/2}}$$

We remark that the term $C(f, \Delta_i)$ depends exponentially on $\Delta_i$, suggesting that $C(f, \Delta_i)$ may be large even when adopting values of $\delta$ that are not arbitrarily close to zero.

Furthermore, let us observe that Algorithm 1 satisfies important properties in the instance-dependent stochastic setting. More precisely, (i) it matches the instance-dependent regret lower-bound, since $f(\cdot)$ can be chosen arbitrarily close to $\log(t)$, (ii) it works without requiring the knowledge of the time horizon $T$, thus being an *any-time algorithm*.

### 3.3 INSTANCE DEPENDENT/INDEPENDENT TRADE-OFF

In this section, we provide a negative result, showing that *no algorithm* can perform well in both the instance-dependent and instance-independent cases, thus suggesting that the two cases need to be studied separately. Initially, we state the following result that relates to the upper regret bounds in the two (instance-dependent/independent) cases.

**Theorem 4** (Instance Dependent/Independent Trade-off). *Let $\pi$ be any policy for the bandits with ranking feedback problem. If $\pi$ satisfies the following properties:*

- *(instance-dependent upper regret bound) $R_T \leq \sum_{i=1}^n C(\Delta_i) T^\alpha$*

- *(instance-independent upper regret bound) $R_T \leq nCT^\beta$*

*then, $2\alpha + \beta \geq 1$, where $\alpha, \beta \geq 0$.*

From Theorem 4, we can easily infer the following impossibility result.

**Corollary 5.** *There is no algorithm for bandits with ranking feedback achieving both subpolynomial regret in the instance-dependent case, i.e., $\forall \alpha > 0, \exists C(\cdot) : R_T \leq \sum_{i=1}^n C(\Delta_i) T^\alpha$, and sublinear regret in the instance-independent case.*

To ease the interpretation of Corollary 5, we discuss the performance of Algorithm 1 in the instance-independent case in the following result.

**Corollary 6.** *For every choice of $\delta > 0$ in $f(t) = \log(t)^{1+\delta}$, there is no value of $\eta > 0$ for which Algorithm 1 achieves an instance-independent regret bound of the form $R_T \leq \mathcal{O}(T^{1-\eta})$.*

The above result shows that Algorithm 1 suffers from linear regret in $T$ in the instance-independent case except for logarithmic terms.

### 3.4 INSTANCE-INDEPENDENT UPPER BOUND

The impossibility result stated by Corollary 5 pushes for the need for an algorithm guaranteeing a sublinear regret in the instance-independent case. Initially, we observe that the standard Explore-then-Commit algorithm (from here on denoted with EC) proposed by Lattimore & Szepesvari (2017) can be applied, achieving a regret bound $\mathcal{O}(T^{2/3})$ in the instance-independent case.

Let us briefly summarize the functioning of the EC algorithm. It divides the time horizon into two phases as follows: (i) *exploration phase*: the arms are pulled uniformly for the first $m \cdot n$ rounds, where $m$ is a parameter of the algorithm one can tune to minimize the regret; (ii) *commitment phase*: the arm maximizing the estimated reward is pulled.

In the case of bandits with ranking feedback, the EC algorithm explores the arms in the first $m \cdot n$ rounds and subsequently pulls the arm in the first position of the ranking feedback received at round $t = m \cdot n$. As is customary in standard (non-ranking) bandits, the best regret bound can be achieved by setting $m = \lceil T^{2/3} \rceil$, thus obtaining $\mathcal{O}(T^{2/3})$.

We show that we can get a regret bound better than that of the EC algorithm. In particular, we provide the Ranking Logarithmic Phased Elimination (R-PLE) algorithm, which breaks the barrier of $\mathcal{O}(T^{2/3})$ guaranteeing a regret $\widetilde{\mathcal{O}}(\sqrt{T})$ when neglecting logarithmic terms. The pseudocode of R-PLE is reported in Algorithm 2.

**R-LPE Algorithm.** In order to proper analyze the algorithm, we need to introduce the two following definitions. Initially, we introduce the definition of the loggrid set as follows,

**Definition 2** (Loggrid). *Given two real numbers $a, b$ s.t $a < b$ and a constant value $T$, we define*

$$LG(a, b, T) := \left\{ \lfloor T^{\lambda_j b + (1 - \lambda_j) a} \rfloor : \ \lambda_j = \frac{j}{\lfloor \log(T) \rfloor}, \ \forall j = 0, \dots, \lfloor \log(T) \rfloor \right\}.$$

Next, we give the notion of active set, which the algorithm employs to cancel out sub-optimal arms.

**Definition 3** (Active set). *We define the active set $\mathcal{F}_t(\zeta)$ at the timestep $t$ of the algorithm, the set of arms*

$$\mathcal{F}_t(\zeta) := \left\{ a \in A : \forall b \in A \ \sum_{\tau = 1 : n | \tau}^{t} \{ \mathcal{R}_\tau(a) > \mathcal{R}_\tau(b) \} \geq \zeta \right\}.$$

*Where the symbol $|$ stands for "divide", so that the condition $\tau | n$ means that we are summing only over the $\tau$ which are multiple of $n$. This condition will be called **filtering condition**.*

---

**Algorithm 2** Ranking Logarithmic Phased Elimination (R-LPE)

---

1: Initialize $S = [n]$
2: Initialize $\mathcal{L} = LG(1/2, 1, T)$
3: **for** $t \in [T]$ **do**
4:     Play $i_t \in \arg\min_{i \in S} Z_i(t)$
5:     Update $Z_i(t)$ number of times $i_t$ has been pulled
6:     Observe ranking $\mathcal{R}_t$
7:     **if** $\min_{i \in S} Z_i(t) \in \mathcal{L}$ **then**
8:         $\alpha = \frac{\log(\min_{i \in S} Z_i(t))}{\log(T)} - \frac{1}{2}$
9:         $S = \mathcal{F}_t(T^{2\alpha})$
10:    **end if**
11: **end for**

---

Initially, we observe that R-LPE differs from Algorithm 1, as it takes into account the whole history of the process and not only the last ranking $\mathcal{R}_t$ received. It also requires the knowledge of $T$.

Set $S$ denotes the active set of arms used by the algorithm. Initially, set $S$ comprises all the possible arms available in the problem (Line 1). Furthermore, the set which drives the update of the decision space $S$, namely $\mathcal{L}$, is initialized as the loggrid built on parameters $1/2, 1, T$ (Line 2).

At every round $t \in [T]$, R-LPE chooses the arm from active set $S$ with the minimum number of pulls, namely $i$ s.t. $Z_i(t)$ is minimized (Line 4); ties are broken by index order. Next, the number of times arm $i_t$ has been pulled, namely $Z_i(t)$, is updated accordingly (Line 5). The peculiarity of the algorithm is that set $S$ changes every time the condition $\min_i Z_i(t) \in \mathcal{L}$ is satisfied (Line 7). When the aforementioned condition is met, the set of active arms $S$ is filtered to avoid the exploration on sub-optimal arms. Precisely, $S$ is filtered given the time dependent parameter $\alpha$ (Line 8- 9).

**Regret Bound.** We state the following theorem providing a regret bound to Algorithm 2 in the instance-independent case.

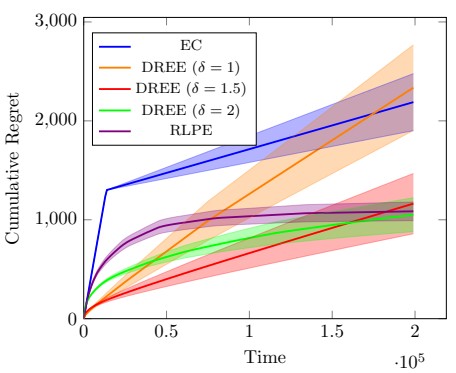 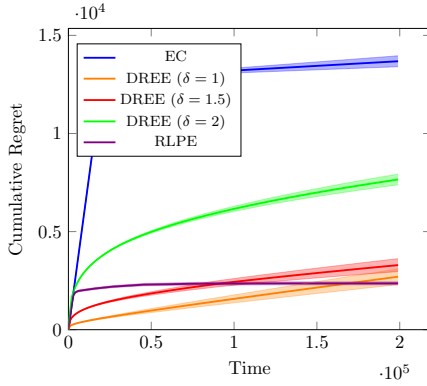

(a) Instance with $\Delta_{\min} = 0.03$ and all the gaps small  (b) Instance with $\Delta_{\min} = 0.03$ and the other gaps big

Figure 1: Cumulative regret for $\Delta_{\min} < 0.05$ (averaged over 50 runs; $95\%$ confidence interval).

**Theorem 7.** *In the stochastic bandits with ranking feedback setting, Algorithm 2 achieves the following regret bound:*

$$R_T \leq \widetilde{\mathcal{O}}\left(n\sqrt{T}\right),$$

*when $n$ arms are available to the learner.*

At first glance, the result presented in Theorem 7 may seem unsurprising. Indeed, there are several elimination algorithms achieving $\mathcal{O}(\sqrt{T})$ regret bounds in different bandit settings (see, for example, (Auer & Ortner, 2010; Lattimore et al., 2020; Li & Scarlett, 2022)). Nevertheless, our setting poses several additional challenges compared to existing ones. For instance, in our framework, it is not possible to rely on concentration bounds, as the current feedback is heavily correlated with the past ones. For such a reason, our analysis employs non-trivial arguments, drawing from recent results in the theory of Brownian Motions, which allow to properly model the particular feedback we propose.

## 4  ANALYSIS IN THE ADVERSARIAL SETTING

We focus on bandits with ranking feedback in adversarial settings. In particular, we show that no algorithm provides sublinear regret without statistical assumptions on the rewards.

**Theorem 8.** *In adversarial bandits with ranking feedback, no algorithm achieves $o(T)$ regret with respect to the best arm in hindsight with a probability of $1 - \epsilon$ for any $\epsilon > 0$.*

*Proof sketch.* The proof introduces three instances in an adversarial setting in a way that no algorithm can achieve sublinear regret in all the three. The main reason behind such a negative result is that ranking feedback obfuscates the value of the rewards so as not to allow the algorithm to distinguish two or more instances where the rewards are non-stationary. The three instances employed in the proof are divided into three phases such that the instances are similar in terms of rewards for the first two phases, while they are extremely different in the third phase. In summary, if the learner receives the same ranking when playing in two instances with different best arms in hindsight, it is not possible to achieve a small regret in both of them. □

## 5  NUMERICAL EVALUATION

This section presents a numerical evaluation of the algorithms proposed in the paper for the *stochastic settings*, namely, DREE and R-LPE. The goal of such a study is to show two crucial results: firstly, the comparison of our algorithms with a well-known bandit baseline, and secondly, the need to develop distinct algorithms tailored for instance-dependent and instance-independent scenarios.

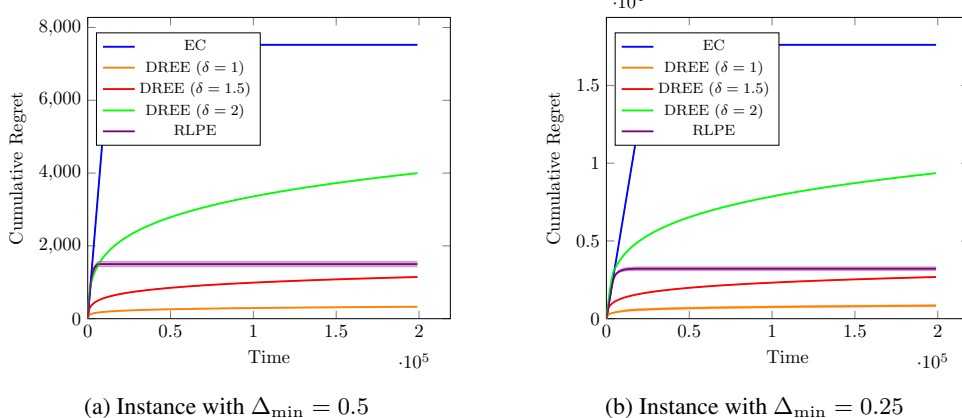

(a) Instance with $\Delta_{\min} = 0.5$         (b) Instance with $\Delta_{\min} = 0.25$

Figure 2: Cumulative regret for $\Delta_{\min} \geq 0.25$ (averaged over 50 runs; 95% confidence interval).

To establish a benchmark for comparison, we consider the EC (Explore-Then-Commit) algorithm, which is one of the most popular algorithms among the explore-then-commit class providing sub-linear regret guarantees. In the following, we evaluate the DREE algorithm with different choices of the $\delta$ parameter in the function $f(t) = \log(t)^{1+\delta}$; precisely, we choose $\delta \in \{1.0, 1.5, 2.0\}$. Furthermore, we consider four stochastic instances whose specific parameters are discussed below. In all these instances, we assume the rewards to be drawn from Gaussian random variables with unit variance, *i.e.*, $\sigma^2 = 1$, and we let the time horizon be equal to $T = 2 \cdot 10^5$. Finally, for each algorithm, we evaluate the cumulative regret averaged over 50 runs.

We structure the presentation of the experimental results into two groups. In the first, the instances have a small $\Delta_{\min}$, while in the second, the instances have a large $\Delta_{\min}$.

**Small Values of $\Delta_{\min}$**  We focus on two instances with $\Delta_{\min} < 0.05$. In the first of these two instances, we consider $n = 4$ arms, and a minimum gap of $\Delta_{\min} = 0.03$. In the second instance, we consider $n = 6$ arms, with $\Delta_{\min} = 0.03$. The expected values of the rewards of each arm are reported in Appendix D, while the experimental results in terms of average cumulative regret are reported in Figures 1a–1b. We observe that in the first instance (see Figure 1a) all the DREE algorithms exhibits a linear regret bound, confirming the strong sensitivity of this family of algorithms on the parameter $\Delta_{\min}$ in terms of regret bound. In contrast, the R-LPE algorithm exhibits better performances in terms of regret bound, as its theoretical guarantee are independent on the values of $\Delta_{\min}$. Furthermore, Figure 1b shows that the DREE algorithms (with $\delta \in 1.0, 1.5$) achieve a better regret bound when the number of arms is increased. Indeed, these regret bounds are comparable to the ones achieved by the R-LPE algorithm. The previous result is reasonable as the presence of $\Delta_i$-s in the regret bound lowers the dependence on the number of arms. It is worth noticing that all our algorithms outperform the baseline EC.

**Large Values of $\Delta_{\min}$**  We focus on two instances with $\Delta_{\min} \geq 0.25$. In the first instance, we consider $n = 4$ arms with a minimum gap of $\Delta_{\min} = 0.5$ among their expected rewards. In the second instance, we instead consider a larger number of arms, specifically $n = 8$, with a minimum gap equal to $\Delta_{\min} = 0.25$. The expected values of the rewards are reported in Appendix D, while the experimental results in terms of average cumulative regret are provided in Figures 2a–2b. As it clear from both Figures 2a–2b when $\Delta_{\min}$ is sufficiently large, the DREE algorithms (with $\delta \in \{1.0, 1.5\}$) achieves better performances with respect both the EC and R-PLE algorithms in terms of cumulative regret. Furthermore, there is empirical evidence that a small $\delta$ guarantees better performance, which is reasonable according to theory. Indeed, when $\delta$ is small, the function $f(t)$, which drives the exploration, is closer to a logarithm. Also, as shown in Corollary 3, when $\Delta_{\min}$ is large enough, the parameter $\delta$ affects the dimension of $C(f, \Delta_i)$ more weakly, which results in a better regret bound.

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
