# OpenReview forum: "Bandits with Ranking Feedback"
_ICLR.cc/2024/Conference — Submitted to ICLR 2024_

### Official Review · Reviewer_hR48 · 2023-10-29

**Soundness:** 3 good
**Presentation:** 3 good
**Contribution:** 3 good
**Rating:** 6
**Confidence:** 3

**Summary:**

In the setting here, playing a bandit arm generates a hidden reward though the decision maker can observe the ordering of the average reward of each arm cumulated up to this point (but not of the empirical rewards themselves). The paper then shows that in the stochastic version of this setting, no algorithm can achieve logarithmic regret. Furthermore, the authors introduce an algorithm coming arbitrarily close to this lower bound (logarithmic regret) and an algorithm achieving regret scaling as $O(\sqrt{T})$ in the instance independent setting. Finally, a few numerical experiments contrast the performance of the algorithm proposed here using the default Explore-and-Commit algorithm as a baseline.

**Strengths:**

Proposes an interesting setting that seems relevant to preference learning in real systems as the only feedback the algorithm receives is a ranking of the unobservable average empirical rewards.

Proposes a method to overcome a non-trivial difficulty posed by this setting (the inability to observe the actual rewards of the arms) and shows the instance-dependent and instance-independent settings need to be treated separately.

Provides optimality proofs for the algorithms and numerical experiments to showcase the proposed algorithms' performance.

The paper is well structures and the writing is clear and benefits from examples of how the mechanics of the setting work.

**Weaknesses:**

I would have liked to see more solid justification of the practical relevance of the setting: an experiment on a real world dataset for instance, and/or a more crystallised example of a real world setting that is accurately modelled by this setting.

I would have liked to see a more detailed description of the Explore-then-Commit implementation used as a baseline. I believe the generic EC algorithm requires a grid as input, what is the choice of this parameter here?

The proposed algorithm (DREE) is not entirely surprising. In light of not knowing neither the optimality gap $\Delta^*$ nor the accumulated rewards, we would not know how tight the confidence interval corresponding to each arm would need to be. Hence, no matter how many times we explore relative to $\log(T)$, we can never be sure we have explored enough. Hence it makes sense that we need to play at a "barely" super-logarithmic rate, in order to ensure we eventually actually discover the best arm (regardless of how small $\Delta^*$ is).

**Questions:**

Regarding the experimental setup:
- Can you offer more details regarding the EC implementation and parameters for the sake of reproducibility?
- DREE($\delta + 1$) performs worst in Figure 1a. but best in the rest of the experiments. Do you have any insights on how would someone implementing your algorithm know in which of the regimes they might find themselves in and how someone can go about choosing the right $\delta$ at design-time?

---

> ### Author Response · Authors · 2023-11-15
>
> > I would have liked to see more solid justification of the practical relevance of the setting: an experiment on a real world dataset for instance, and/or a more crystallised example of a real world setting that is accurately modelled by this setting.
>
> In our formal model, a learner can distinguish which arm is the best given a number of observations, but they cannot assign scores to the arms and therefore they cannot evaluate how much an arm is better than the others. In principle, every time the reward cannot be easily measurable by a user, e.g. when the reward is not a monetary amount nor a probability conversion nor a number, ranking bandits provide a natural solution. Imagine a matchmaking problem among people. In this case, it is not obvious for a person to score other people. Given the increasing attention in the scientific literature for human-centered settings where humans will interact with algorithms more and more frequently and given that humans may have difficulties in scoring items, we believe that our setting may play a central role. Furthermore, in some applications, there can be a mediator, and a user can prefer not to reveal the scores as these are sensitive data for the company. In the paper, we provide the advertising example in which companies usually do not communicate to the advertising platforms the value per clicks of their ad campaigns. We will clarify better these arguments in the paper.
>
> Let us remark that our model can be enriched along several directions, reducing more and more the gap with real-world applications. For instance, we intend to introduce a tolerance such that the learner cannot distinguish among several rankings when the rewards provided by, e.g., two arms are very close. However, any possible extension of the model does not make sense without an accurate study of the basic model.
>
> > I would have liked to see a more detailed description of the Explore-then-Commit implementation used as a baseline. I believe the generic EC algorithm requires a grid as input, what is the choice of this parameter here? Can you offer more details regarding the EC implementation and parameters for the sake of reproducibility?
>
> The Explore and Commit algorithm (Lattimore e Szepesvári - 2020) only needs a parameter $m$ as input, which is the number of times each arm has to be explored. When $\Delta^*$ is unknown, the optimal choice of $m$ is $T^{2/3}$, which is the one chosen in our paper. We will surely enlarge the description of the explore and commit algorithm in the final version of the paper.
>
>
> >The proposed algorithm (DREE) is not entirely surprising. In light of not knowing neither the optimality gap $\Delta^*$ nor the accumulated rewards, we would not know how tight the confidence interval corresponding to each arm would need to be. Hence, no matter how many times we explore relative to $\log(T)$, we can never be sure we have explored enough. Hence it makes sense that we need to play at a "barely" super-logarithmic rate, in order to ensure we eventually actually discover the best arm (regardless of how small $\Delta^*$ is).
>
> We agree with the Reviewer that the intuition behind the Dree algorithm is clear. Nevertheless, please notice that: i) this is the first result in the bandits literature where a superlogarithmic dependence of $T$ is optimal, thus, the result is indeed novel; ii) the paper presents many theoretical upper-bound and lower-bound resorting to theoretical tools that range from classical online learning techniques to Brownian motions, thus, the set of results presented are indeed novels and highly non-trivial.
>
>
> >DREE($\delta$+1) performs worst in Figure 1a. but best in the rest of the experiments. Do you have any insights on how would someone implementing your algorithm know in which of the regimes they might find themselves in and how someone can go about choosing the right $\delta$ at design-time?
>
> This parameter defines the exploration: the higher $\delta$, the more exploration steps are done. As in the well-known $\varepsilon-$greedy algorithm, an higher value of $\varepsilon$ should be used when there are arms with small gaps, the same holds for the parameter $\delta$. Not by chance, Figure 1a which you were mentioning represents the experiment where the gaps are smaller, so that more exploration is needed.

---

> > ### Comment · Reviewer_hR48 · 2023-11-20
> > **Thank you for the clarifications**
> >
> > Thank you for your clarifications, I will keep them in mind during the discussion phase.

---

### Official Review · Reviewer_qw3s · 2023-11-01

**Soundness:** 2 fair
**Presentation:** 3 good
**Contribution:** 2 fair
**Rating:** 3
**Confidence:** 3

**Summary:**

Unlike traditional bandits or dueling bandits, the authors introduce a variation of ranking based feedback where the learner observes a continuously updated leaderboard of arms, without direct numerical feedback. They study the problem of designing noregret algorithms with ranking feedback both in the stochastic and adversarial settings, showing certain algorithmic theorems do largely transfer over.

**Strengths:**

The authors do a good job of exhaustively considering the most relevant settings of stochastic vs adversarial, instance dependent vs independent, providing interesting upper and lower bounds, with comparison to the regular bandit setting. The authors do a relatively clear job of presenting their novel setting and give some insights into previous works and how their setting differentiates from previous works.

**Weaknesses:**

The main weaknesses of the paper stem from the lack of theoretical intuition and understanding of the main proof results. For example, the instance-dependent lower bound presented in Theorem 1 hinges on the separation lemma in Lemma 1, which only shows that there is some positive probability that the ranking leaderboard never changes. The fact that there is a positive probability that a possible event occurs is not surprising, as such a probability could be arbitrarily small. Therefore, it is extremely unclear why in Theorem 1, the regret will need to be substantially larger than the stated bounds as such a small-probability event may only introduce Omega(1) regret into the expectation. Note that most lower bounds use Omega(*) notation, which means the lower bounds should hold up to multiplicative constants. The proof sketches from other theorems generally suffer from a similar lack of intuition, clarity, and possibly correctness. Generally it is surprising that a sqrt(T) instance-independent bound could be derived from such weak bounds given in Theorem 2, which has a exponential dependence on 1/Delta_i.

Furthermore, the setting is rather arbitrary and there are no clear downstream applications that would utilize such a specific feedback model. It is also unclear why the dueling bandit setting is insufficient for dealing with preference feedback and while there are extensive comparisons made with the typical bandit setting, it is unclear how the stated theoretical results would compare with theoretical results in the dueling bandit setting, which is the most relevant setting.

**Questions:**

In Lemma 1, what are the concrete bounds on the probability of the bad event occuring?

In theorem 1, how does such bad events translate to significantly higher regret?

In theorem 2, are the exponential bounds necessary (a lower bound perhaps?) and how are you able to derive instance-independent regret from this?

Lastly, what is one specific ML application that would benefit from this setting vs dueling bandits? How do your theoretical bounds in the instance-dependent and independent settings differ from similar results in the dueling bandits case?

---

> ### Author Response · Authors · 2023-11-15
>
> First of all we want to thank the Reviewer for the numerous feedback which will allow us to clarify crucial points of our paper. Let us remark that we devote the first part of our rebuttal to rebate the Reviewer's statement about the incorrectness of our theoretical results. **Since we are confident about the correctness and the novelty of our results**, please let us know if the following arguments are clear enough. Then, in the second part of the rebuttal, we will answer the questions related to settings and curiosity.
>
> > The main weaknesses of the paper stem from the lack of theoretical intuition and understanding of the main proof results. For example, the instance-dependent lower bound presented in Theorem 1 hinges on the separation lemma in Lemma 1, which only shows that there is some positive probability that the ranking leaderboard never changes. The fact that there is a positive probability that a possible event occurs is not surprising, as such a probability could be arbitrarily small. Therefore, it is extremely unclear why in Theorem 1, the regret will need to be substantially larger than the stated bounds as such a small-probability event may only introduce Omega(1) regret into the expectation. Note that most lower bounds use Omega(*) notation, which means the lower bounds should hold up to multiplicative constants.
>
> We observe that the separation lemma presented in Lemma 1 is formulated in a way that the probability holds simultaneously for all the pairs $t,t' \in \mathbb{N}^*$, i.e.,
> $$ \mathbb P\Big (\forall t, t' \in \mathbb{N}^* G_t/t\ge G'_{t'}/{t'}\Big )>0.$$
> In general, to employ a "high-probability" argument to bound the cumulative regret, it is necessary to set the probability of the "bad event" as a function of the time horizon $T$ (for example, equal to $\mathcal O(T^{-1})$ or at least $\mathcal O(T^{-1/2})$). Thus, it is incorrect to claim that such an event is a low-probability event, as its probability is fixed and independent of the time horizon $T$.
>
> >The proof sketches from other theorems generally suffer from a similar lack of intuition, clarity, and possibly correctness. Generally, it is surprising that a $\sqrt(T)$ instance-independent bound could be derived from such weak bounds given in Theorem 2, which has an exponential dependence on $1/\Delta_i$.
>
> We notice that DREE (Algorithm 1), which is the algorithm satisfying theorem 2, is not guaranteed to achieve an instance independent regret of $\sqrt T$, as shown in Corollary 6. Indeed, the algorithm achieving an instance independent regret of $\sqrt T$ is R-LPE (Algorithm 2). This is because the crucial feature characterising our setting is that any algorithm cannot achieve both good instance-dependent and instance-independent regret bound (see Theorem 4).
>
> >In theorem 2, are the exponential bounds necessary (a lower bound perhaps?) and how are you able to derive instance-independent regret from this?
>
> The Reviewer is correct. The exponential dependence in $1/\Delta_i$ is necessary to achieve a poly-logarithmic instance-dependent regret bound. Indeed, if it were possible to achieve an instance **dependent** regret bound of the form:
> $$R_T\le C\Delta^{-\alpha}P(T),$$
> for a function $P(T)$ which is polylogarithmic in $T$, it would imply the following bound for the instance \textbf{independent} regret:
>  $$R_T\le \sup_{\Delta>0} \min\{C\Delta^{-\alpha}P(T), \Delta T\}.$$
> Notice that, for $\Delta \ge T^{-1/(\alpha+1)}$ the first term is less than $CT^{\frac{\alpha}{\alpha+1}}P(T)$, while for $\Delta \le T^{-1/(\alpha+1)}$ the second one is less than $T^{\frac{\alpha}{\alpha+1}}$. Therefore, the full instance-independent regret would be bounded by
> $$R_T\le CT^{\frac{\alpha}{\alpha+1}}P(T),$$
> which contradicts Corollary 5. Therefore, the exponential (or at least superpolynomial) dependence on $(1/\Delta)$ is unavoidable. We can add this derivation to the appendix.
>
> > In Lemma 1, what are the concrete bounds on the probability of the bad event occurring?
>
> Answer previously.
>
> >In theorem 1, how does such bad events translate to significantly higher regret?
>
>  Answer previously.

---

> ### Author Response · Authors · 2023-11-15
>
> >Furthermore, the setting is rather arbitrary and there are no clear downstream applications that would utilize such a specific feedback model. It is also unclear why the dueling bandit setting is insufficient for dealing with preference feedback and while there are extensive comparisons made with the typical bandit setting, it is unclear how the stated theoretical results would compare with theoretical results in the dueling bandit setting, which is the most relevant setting.
>
> **From an application perspective**, in our formal model, a learner can distinguish which arm is the best given a number of observations, but they cannot assign scores to the arms and therefore they cannot evaluate how much an arm is better than the others. In principle, every time the reward cannot be easily measurable by a user, e.g. when the reward is not a monetary amount nor a probability conversion nor a number, ranking bandits provide a natural solution. Imagine a matchmaking problem among people. In this case, it is not obvious for a person to score other people. Given the increasing attention in the scientific literature for human-centered settings where humans will interact with algorithms more and more frequently and given that humans may have difficulties in scoring items, we believe that our setting may play a central role. Furthermore, in some applications, there can be a mediator, and a user can prefer not to reveal the scores as these are sensitive data for the company. In the paper, we provide the advertising example in which companies usually do not communicate to the advertising platforms the value per clicks of their ad campaigns. We will clarify better these arguments in the paper.
>
> Let us remark that our model can be enriched along several directions, reducing more and more the gap with real-world applications. For instance, we intend to introduce a tolerance such that the learner cannot distinguish among several rankings when the rewards provided by, e.g., two arms are very close. However, any possible extension of the model does not make sense without an accurate study of the basic model.
>
> **From a theoretical perspective**, in dueling bandits, similar performances to standard multi-armed bandits problem are achievable, namely, $\log(T)$ in the stochastic setting, $\sqrt{T}$ in the adversarial one. Nevertheless, the regret definition between dueling bandits and multi-armed bandits is **different**; indeed, in dueling bandits the regret is computed w.r.t. the arm which may win more duels, while in multi-armed bandits the regret is computed w.r.t. the arm with the larger mean (similarly to bandits with ranking feedback).
>
>
> >Lastly, what is one specific ML application that would benefit from this setting vs dueling bandits? How do your theoretical bounds in the instance-dependent and independent settings differ from similar results in the dueling bandits case?
>
> Answer previously.

---

> ### Comment · Reviewer_qw3s · 2023-11-17
>
> Thanks for the rebuttal but I'm still unsure about the correctness and validity of the setting. Let's focus on the following two specific concerns for now.
>
> For theory: In your separation lemma, you showed that the probability that some possible event (namely that the ranking never switches) occurs is positive. But is that not a statement that is trivially true: any possible outcome has positive probability? Indeed, the probability could be extremely small, like $e^{-100T}$ and so how can you prove a lower bound that hinges on a low probability event?
>
> For application: You mentioned that "As mentioned, the regret definition between dueling bandits and multi-armed bandits is different; indeed, in dueling bandits the regret is computed w.r.t. the arm which may win more duels, while in multi-armed bandits the regret is computed w.r.t. the arm with the larger mean (similarly to bandits with ranking feedback)."
>
> Actually, I believe the novelty in your case is not the regret definition but the feedback being a cumulative ranking feedback, which I think leads to more difficult inference. My question is: why is this cumulative ranking feedback useful to consider when in all the applications you described, one would receive instantaneous non-cumulative ranking feedback?

---

> > ### Author Response · Authors · 2023-11-19
> >
> > > For theory.
> >
> > We believe that there is a possible misunderstanding on this crucial aspect of our work. The probability that the aforementioned event holds *does not depend on the time horizon T*. Indeed, the probability of this event depends only on the gap between the arms $\Delta$, namely, the difference between the drifts of the random walks. Thus, for any value of the time horizon $T>0$, this probability can be much larger compared to the value of $e^{-100T}$. Furthermore, by letting $p(\Delta)$ the probability that such an event holds (since it depends only on $\Delta$ and not on $T$), if we ignore $p(\Delta)$ from our analysis as suggested by the Reviewer, we could incur an instance-dependent regret of the order $p(\Delta)T+...$, which is linear in $T$. To conclude, we observe that the Reviewer's statement "you prove a lower bound that hinges on a low probability event" is not correct. Our lower bound relies on a fixed probability event that only depends on the gap $\Delta$ between the arms.
> >
> >
> > >For application.
> >
> > The Reviewer is correct when they state that our feedback is a form of "cumulative ranking feedback”; indeed, the ranking feedback also depends on the history of the previously selected arms. Nevertheless, in the advertising example presented in the paper, we discuss a scenario where an automatic bidding platform, without direct access to the actual revenue of advertising campaigns, only observes a score describing how advertisers have performed in the past in terms of revenue. In such an example, the ranking of the score clearly depends on the previous advertisers' performances; thus, it depends on the history of the process. As a final remark, we thank the Reviewer for the observation and, in the final version of the paper, we will extend the latter example to better emphasize the fact that the ranking feedback depends on the history of the process.

---

> ### Comment · Reviewer_qw3s · 2023-11-21
>
> Thanks for the feedback. I generally think that ranking feedback can have some applications but do not believe this paper has concrete mathematical insights. For example, in the first lemma, the authors mentioned that "probability of this event depends only on the gap between the arms, namely, the difference between the drifts of the random walks". However, all that was shown in the theorem statement was that the probability is positive. This clearly needs more elucidation, especially what happens when the gap is 0 or extremely large.

---

> > ### Author Response · Authors · 2023-11-22
> >
> > In the statement of Lemma 1 we just affirm that the probability of that event is positive because it is the only property we need to derive Theorem 1. Thus, we did not provide a full computation of such an event in the paper. Moreover, asking whether the probability of an event of the form $\forall t\in \mathbb N^*, \dots $ depends on $t$ is like asking if the result of $\sum_{n=1}^\infty \frac{1}{n^2}$ depends on $n$: it is not something that should need an explanation. Nevertheless, the probability that the aforementioned event holds can be computed by means of Lemma 4 and Lemma 1.

---

### Official Review · Reviewer_7Kt2 · 2023-11-01

**Soundness:** 3 good
**Presentation:** 3 good
**Contribution:** 3 good
**Rating:** 8
**Confidence:** 4

**Summary:**

In this paper, the authors study setting of learning with bandit feedback where the learner gets to see only the *ranking* between different arms based on the average rewards received by each arm so far. In the stochastic case, where the rewards of each arm follow a fixed distribution, the authors show instance dependent lower bounds (that is, lower bounds that depend on $\mu_i^* - \mu_i$, where $\mu_i$ is the mean reward of arm $i$ and $i^*$ is the arm with highest mean rewards)  and algorithms that enjoy regret guarantees closely matching the lower bounds. The authors also show an interesting trade-off between instance-dependent and instance-independent guarantees for any bandit algorithm based only on ranking feedback, and provide a bandit algorithm with instance-independent guarantees. Finally, they conclude showing that in the adversarial setting no algorithm can guarantee sublinear regret and with a few numerical experiments.

**Strengths:**

- The setting is natural (although the motivation might be debatable, as I mention in the weaknesses) and quite interesting,  since it clearly requires different kinds of techniques from traditional bandit algorithms. The results are elegant and the investigation the authors perform goes over many of the natural questions one could consider (upper and lower bounds that nearly match, trade-off between instance dependency/independency, etc);
- The presentation does a good job of describing the results and main algorithm in the main paper (although, saddly, not much discussion about the proof techniques, but you can only do so much in a few pages);
- I have read the first couple of proofs in the paper, and they are clear and not too hard to follow;

**Weaknesses:**

- One of the motivations in the introduction of the paper are cases when the rewards are not (easily) representable by numerical values. However, the setting in the end still relies on the existence of numerical values for the rewards, but the learner cannot see them. This might not be a weakness of the paper, but that is something that when reading the introduction I did not expect and goes without being deeply discussed in the paper (is this the case with dueling bandits as well?). I still think the setting is interesting, but I am not sure if the motivation in the introduction correctly matches the setting. Although I cannot easily see practical applications, the theoretical framework is still super interesting (i.e., algorithms that only get information about the ranking of the received rewards), the analysis are elegant, and I can be easily be proven wrong and follow up work can have interesting applications of this setting. This discussion seems interesting, and in the questions section I have a related question that could be interesting to incorporate in the final version of the paper (if the authors agree that it is an interesting point, of course);
- The experiments are very small in scale. They do convey a message (the difference in empirical performance for algorithms that focus on the instance-dependent guarantees and the instance-independent guarantees), but since the algorithms are already implemented, it would have been interesting to see their performance in cases with more arms. I am not sure if there are computational constraints that prevent experimentation with more arms. Moreover, I think the main paper has the wrong plots (that are fixed in the supplementary material version), right? This could have been made explicit in notes in the appendix to not make the reviewers confused.

**Questions:**

- In the paper, the guarantees of Algorithm 1 seem to require the distribution of the rewards of each arm to be subgaussian, while this assumption seems absent from the guarantees of Algorithm 2. Is this right? So it is the case that algorithm 2 would preserve its guarantees even with heavier tailed (say, sub-exponential) rewards while this would not necessarily be true for algorithm 1?
- Just to triple check, the plots in the main paper are not the right ones, right? The plots in the supplementary seem to be very different and to actually convey the message the authors meant to convey.
- Is the "full-information with ranking feedback" easy? It seems that even with full information (that is, we get to rewards of all arms at each round, but only see the ranking) also seems not trivial, but I might be mistaken. If the authors have not thought too much about this, don't worry;
- This is a more complicated question, and the authors should feel free to not answer this in the rebuttal if they are time constrained. But one thing that I noticed in the case with ranking feedback is that it is somewhat weaker than in the case of dueling bandits, even though ranking feedback requires a total order of the arms at every round while dueling bandits do not (to the best of my knowledge). Yet, we cannot use dueling bandit algorithms directly in this setting, since we only see the entire ranking based on the *cumulative reward* while dueling bandits make comparisons of the *instantaneous rewards* of two arms (if I am not mistaken). However, if we could compare the arms pulled in subsequent rounds, we can could use dueling bandits algorithms. Would this more powerful feedback yield stronger regret guarantees by using dueling bandits? Maybe the crux of the question is: if we were to compare the regret bounds achievable for dueling bandits vs the ones achieved by ranking feedback, are the ones with ranking feedback worse? If so, would this augmented feedback make ranking feedback boil down to dueling bandits?

---

> ### Author Response · Authors · 2023-11-15
>
> > One of the motivations in the introduction of the paper are cases when the rewards are not (easily) representable by numerical values. However, the setting in the end still relies on the existence of numerical values for the rewards, but the learner cannot see them. This might not be a weakness of the paper, but that is something that when reading the introduction I did not expect and goes without being deeply discussed in the paper (is this the case with dueling bandits as well?). I still think the setting is interesting, but I am not sure if the motivation in the introduction correctly matches the setting. Although I cannot easily see practical applications, the theoretical framework is still super interesting (i.e., algorithms that only get information about the ranking of the received rewards), the analysis are elegant, and I can be easily be proven wrong and follow up work can have interesting applications of this setting. This discussion seems interesting, and in the questions section I have a related question that could be interesting to incorporate in the final version of the paper (if the authors agree that it is an interesting point, of course);
>
> We agree with the Reviewer that our setting mainly poses (and, we believe, answers) many interesting **theoretical questions**. Nevertheless, we believe many real-world scenarios could benefit from our work or its possible future extension. In practice, our work applies every time a learner cannot assign scores to arms. While ours is not the case of settings with monetary values or conversion probabilities or numbers (of sales, users, or others), ours captures the natural case in which a human is asked to provide a valuation over other items/people according to non-directly measurable criteria. For instance, it is not easy for a human to assess the value of another person in a matchmaking setting. Given the increasing attention in the scientific literature for human-centered settings where humans will interact with algorithms more and more frequently and given that humans may have difficulties in scoring items, we believe that our setting can play a central role. However, we agree with the Reviewer that we need to clarify better this argument in the paper.
>
> >The experiments are very small in scale. They do convey a message (the difference in empirical performance for algorithms that focus on the instance-dependent guarantees and the instance-independent guarantees), but since the algorithms are already implemented, it would have been interesting to see their performance in cases with more arms. I am not sure if there are computational constraints that prevent experimentation with more arms. Moreover, I think the main paper has the wrong plots (that are fixed in the supplementary material version), right? This could have been made explicit in notes in the appendix not to confuse the reviewers.
>
> We agree with the Reviewer that plots with a relatively small number of arms may surprise the reader. Nevertheless, please notice that the message we wanted to convey mainly depends on the value $\Delta^*$; thus, a larger number of arms with the same $\Delta^*$ would lead to similar plots with a different regret scale, not being of particular interest.
> Please see the fourth question for the correctness of the plots.
>
>
> >In the paper, the guarantees of Algorithm 1 seem to require the distribution of the rewards of each arm to be subgaussian, while this assumption seems absent from the guarantees of Algorithm 2. Is this right? So it is the case that algorithm 2 would preserve its guarantees even with heavier tailed (say, sub-exponential) rewards while this would not necessarily be true for algorithm 1?
>
> We thank the Reviewer for having pointed out the typo. Even the second algorithm does not work without assuming the subgaussianity of the distribution. We will surely correct the typo in the final version of the paper.
>
>
> >Just to triple check, the plots in the main paper are not the right ones, right? The plots in the supplementary seem to be very different and to actually convey the message the authors meant to convey.
>
> All the plots should be correct. The difference between the plots in the main papers and the ones in the supplementary is that, in the appendix, the regret is normalized for the theoretical upper bound; thus, the functions converge to a constant. On the contrary, in the main paper, the regret is plotted cumulatively.

---

> ### Author Response · Authors · 2023-11-15
>
> >Is the "full-information with ranking feedback" easy? It seems that even with full information (that is, we get to rewards of all arms at each round, but only see the ranking) also seems not trivial, but I might be mistaken. If the authors have not thought too much about this, don't worry;
>
> We thank the Reviewer for the interesting question. Please notice that in the full-information with ranking feedback setting, the learner would observe the ranking of the empirical mean of the rewards computed by sampling the rewards for **every** arm at each round. Thus, the learner would have access to an unbiased estimator of the ranking, which, in the bandit setting, would be possible only by sampling every arm uniformly at random (pure exploration). Thus, in the stochastic setting, the optimal algorithm would be trivial, namely, always playing the arm that is ranked first, leading to constant regret. In the adversarial setting, we conjecture that achieving $\sqrt{T}$ would be possible (in the full-feedback setting) simply by employing regularization techniques.
>
> >This is a more complicated question, and the authors should feel free to not answer this in the rebuttal if they are time constrained. But one thing that I noticed in the case with ranking feedback is that it is somewhat weaker than in the case of dueling bandits, even though ranking feedback requires a total order of the arms at every round while dueling bandits do not (to the best of my knowledge). Yet, we cannot use dueling bandit algorithms directly in this setting, since we only see the entire ranking based on the cumulative reward while dueling bandits make comparisons of the instantaneous rewards of two arms (if I am not mistaken). However, if we could compare the arms pulled in subsequent rounds, we can could use dueling bandits algorithms. Would this more powerful feedback yield stronger regret guarantees by using dueling bandits? Maybe the crux of the question is: if we were to compare the regret bounds achievable for dueling bandits vs the ones achieved by ranking feedback, are the ones with ranking feedback worse? If so, would this augmented feedback make ranking feedback boil down to dueling bandits?
>
> We thank the Reviewer for the interesting question. The intuition of the Reviewer is correct; dueling bandits present a different feedback with respect to bandits with ranking feedback, since in bandits with ranking feedback, the ranking of **empirical means** is observed. Nevertheless, assuming that the feedback types were somehow comparable, we underline that the regret definition of the proposed setting is different.  Precisely, dueling bandit aims to find the arms that would win more duels, while, in bandits with ranking feedback, the aim is to find the arm with the larger mean (when the variance is large, the arm with larger mean is not the one who wins more duel). Thus, the regret is defined accordingly, leading to a different regret definition.

---

> > ### Comment · Reviewer_7Kt2 · 2023-11-23
> > **Great replies and overall thoughts**
> >
> > I would like to thank the authors for providing thorough replies to my concerns and questions (even to those that were more out of curiosity). The small discussion on the full information setting is a nice thought experiment that could maybe be mentioned in the paper if space allows. I am sorry for not engaging during the discussion period, but I had to focus my time to borderline papers, while this one I was comfortable with my score even after looking at the rebuttals and other reviews. Just a quick comments on the plots: what I meant is that the plots on the pdf given as main paper and the one in the supplementary material are a bit different, be sure to pick the right one for the final version of the paper.
> >
> > Looking at the other reviews, it seems that I am the most positive reviewer. In summary, the other reviewers were concerned about the practicality of the setting and on the correctness and/or intuition of the proofs.
> >
> > - On the practicality of the setting, although I agree that this is not clear, I do not think it is crucial for the main message of the paper. The theoretical setting is interesting on its own and the proof techniques are interesting. Having a clearer application would certainly make the paper stronger, but I do not agree that the lack of direct practical applications should be a reason for rejection;
> >
> > - On the correctness, it is hard for us as conference reviewers to assess correctness. It seems that a few of the problems were regarding clarity of the results. Based on the reviews, I'd suggest the authors maybe discuss some of the subtleties on your results (maybe even in the appendix due to space constraints). For example, one aspect that came up in the reviewer discussion period was the order of quantifiers in the results and its importance for the interpretation of the theorems (such as the function $C$ in many of the bounds). So incorporating some of this discussion to prevent misunderstandings like these would be helpful.
> >
> > Overall, I still maintain my opinion that this papers seems correct and that the setting and results are of interest for the community working on the theory of bandit algorithms.

---

### Official Review · Reviewer_qbW7 · 2023-11-01

**Soundness:** 2 fair
**Presentation:** 2 fair
**Contribution:** 2 fair
**Rating:** 3
**Confidence:** 3

**Summary:**

This paper considers bandit with unobservable rewards, instead, the learner can only observe the arm ranking as feedback which is based on their cumulative rewards. The paper shows an instance-dependent regret lower bound that indicates no algorithm can suffer a logarithmic regret over time. The paper then presents two policies, one achieves instance-dependent regret upper bound matching their lower bound, the other achieves instance-independent regret of $\tilde{O}(\sqrt{T})$. They also show that no algorithm can have an optimal regret bound in both instance-dependent and instance-independent cases.

**Strengths:**

* The problem is well formulated, and analyzed in details. Specifically, the regret is analyzed in both instance-dependent and instance-independent cases, as well as their lower bounds.
* The model has practical use cases, e.g., when dealing with human preferences such as in matchmaking settings among humans and when the scores cannot be revealed for privacy or security reasons.

**Weaknesses:**

* The two algorithms are both based on explore-then-commit, which is well-known for its suboptimality on regret. There are other algorithms that have been shown optimal in regret and do not require an estimate on arm empirical means, e.g., another famous one in bandit literature $\epsilon$-greedy. I recommend considering other algorithms that could potentially improve the performance, or discuss why explore-then-commit performs better than $\epsilon$-greedy in this problem.
* The problem formulation is interesting, but the paper organization can still be improved. There could be more discussion on novel techniques used in the proofs and novel algorithm designs to help readers understand the contributions, and some well-known content of bandit problem can be less mentioned.

**Questions:**

* In Theorem 1, I have no clue why there is a function $C(\cdot)$ in the lower bound that has unknown shape and unknown parameter. Without a clear definition of such function, the lower bound seems to have very little meaning to me.
* The description of Algorithm 1 is inaccurate: according to the description, when $t\leq n$, the algorithm plays two arms at each round.
* In Theorem 4, it is surprising to me that there exists tradeoff between instance dependent and independent regret bound, since in most cases an optimal policy has both regret bounds being optimal. I would recommend providing discussion about how to understand such tradeoff.
* I cannot go through all the proofs, but the proof of Theorem 4 seems insufficient. The theorem is for general bandit with ranking feedback problem, however, the proof only shows the constructed two-arm instance but does not extend to general instance.
* In Definition 3, is it feasible to compare the sum of sets to a constant?
* I can hardly understand the basic idea of Algorithm 2, especially why defining active set like that and how it works. I recommend providing discussion about the algorithm design idea and how such design works to make the algorithm finally converge to the optimal action.
* Some of the plots are covered by the legends.

---

> ### Author Response · Authors · 2023-11-15
>
> First of all we want to thank the Reviewer for the numerous feedback which will allow us to clarify crucial points of our paper. Let us remark that we devote the first part of our rebuttal to rebate the Reviewer's statement about the incorrectness of our theoretical results. **Since we are confident about the correctness and the novelty of our results**, please let us know if the following arguments are clear enough. Then, in the second part of the rebuttal, we will answer the questions related to settings and curiosity.
>
> >The two algorithms are both based on explore-then-commit, which is well-known for its suboptimality on regret. There are other algorithms that have been shown optimal in regret and do not require an estimate on arm empirical means, e.g., another famous one in bandit literature $\epsilon$-greedy. I recommend considering other algorithms that could potentially improve the performance or discuss why explore-then-commit performs better than $\epsilon$-greedy in this problem.
>
> We thank the Reviewer for the observation. Nevertheless, we notice that both the DREE and the R-LPE algorithms achieve optimal performance with respect to the lower bounds presented in the paper. Thus, no better algorithms can be designed. Furthermore, we remark that our algorithms present important novelties w.r.t. the classical explore-and-commit approach, which prescribes the learner to explore the arms in the initial rounds and then commit to the one achieving better performances. Our DREE algorithm does not prescribe the learner to uniformly explore the arms in the first $m$ rounds. Instead, the exploration phase continues throughout the entire learner-environment interaction as long as the number of times an arm has been pulled is smaller than $f(t)$. Our R-LPE is also completely different, as it not only considers the current position of an arm in the ranking, but it also takes into account how many times an arm was ranked before any given other arm.
>
> >The problem formulation is interesting, but the paper organization can still be improved. There could be more discussion on novel techniques used in the proofs and novel algorithm designs to help readers understand the contributions, and some well-known content of bandit problem can be less mentioned.
>
> We thank the Reviewer for insightful comment. In the final version of the paper, we will surely enlarge the description of the required mathematical tools since they may not be commonly employed in the bandit literature. Nevertheless, we notice that the necessary mathematical background for a complete understanding of the technical aspects of our work is extensive. Due to lack of space, we were unable to include all the details in the main paper.
>
> > In Theorem 1, I have no clue why there is a function C in the lower bound that has unknown shape and unknown parameter. Without a clear definition of such function, the lower bound seems to have very little meaning to me.
>
>
> The function $C: [0, \infty ) \to [0, \infty )$ does not have a fixed shape a priori, as fixing the shape of such a function may hinder the generality of Theorem 1. More specifically, for any possible choice of the function $C: [0, \infty ) \to [0, \infty )$, there always exists an instance (a collection of $\Delta_i$) such that the regret of any algorithm is $R_T > C(\Delta_i) \log(T)$. In this way, we rule out the possibility for any algorithm to achieve an instance-dependent regret bound which grows logarithmically in the time horizon $T$.
>
> >In Theorem 4, it is surprising to me that there exists tradeoff between instance dependent and independent regret bound, since in most cases an optimal policy has both regret bounds being optimal. I would recommend providing discussion about how to understand such tradeoff.
>
> This represents a crucial point in our paper, and we thank the Reviewer for the opportunity to better explain it. If a policy achieves good performance in terms of instance-dependent regret bound (with respect to the dependence on T), it is possible to find two instances and a gap between the arms, depending on the time horizon, such that any algorithm is prevented from achieving good performance in terms of instance-independent regret bound. The reason behind that lies in the fact that an algorithm achieving a good instance-dependent regret bound must pull the last arm in the ranking very few times. Such a choice, in the instance-independent case, may represent an issue **as the gap between arms is allowed to depend on $T$**. This is because, by means of the separation lemma (Lemma 1), we know that, counterintuitively, the optimal arm may end up being last in the ranking very often in the first steps. Therefore, if an algorithm pulls the last arm in the ranking only a few times, it cannot find the optimal arm.
>
> Since this aspect is crucial, please let us know if the discussion is not clear enough.

---

> ### Author Response · Authors · 2023-11-15
>
> >I cannot go through all the proofs, but the proof of Theorem 4 seems insufficient. The theorem is for general bandits with ranking feedback problems. However, the proof only shows the constructed two-arm instance but does not extend to general instances.
>
> By assumption, the policy  satisfies the following properties:
>
> - (instance-dependent upper regret bound)
> $R_T\le \sum_{i=1}^n C(\Delta_i)T^{\alpha}$
> - (instance-independent upper regret bound)
> $R_T\le nCT^{\beta}$
>
> This is true, particularly for $n=2$ over which we build our proof. This may seem confusing, but the point is that the thesis ($2\alpha + \beta \ge 1$) does not concern the dependence of the regret on the number of actions $n$, but just the one on the time horizon. Therefore, having assumed something else like $R_T\le \sqrt nCT^{\beta}$ or $R_T\le e^nCT^{\beta}$ would make no difference.
>
> > I can hardly understand the basic idea of Algorithm 2, especially why defining active set like that and how it works. I recommend providing discussion about the algorithm design idea and how such design works to make the algorithm finally converge to the optimal action.
>
> We agree with the Reviewer that, due to lack of space, the intuition behind Algorithm 2 is not provided properly. Nevertheless, please notice that the mathematical background required to the algorithms presented in the paper is quite original for the bandit literature. Thus, the intuition behind them is not easy to explain in such a short amount of space. The idea is the following: we start pulling all the arms in parallel, so that we can determine how many times the sample mean of one is larger than the sample mean of the other.
>
> This amount corresponds to the time spent by a random walk with drift equal to the difference between the means of the two arms in the interval $[0,+\infty)$. Since the distribution of this random variable is somehow determined by Levy's arcsin law and related theorems, we can use it to gather statistical evidence for arms that are not optimal. Discarding arms that are probably non optimal according to this condition allows to deal with the exploration-exploitation dilemma. Nevertheless, we will surely enlarge the description of the algorithm in the final version of the paper.
>
> > The description of Algorithm 1 is inaccurate: according to the description, when $t \le n$ the algorithm plays two arms at each round.
>
> We thank the Reviewer for the observation. In Algorithm 1, at line 4, there is a typo where we wrote "end if" instead of "else if".
>
> >In Definition 3, is it feasible to compare the sum of sets to a constant?
>
> Given $a,b \in A$, we denote with the notation $ \\\{ R_t (a) > R_t(b) \\\} $ the indicator function of such an event. This means that $\\\{ R_t (a) > R_t(b) \\\} =1$ if $R_t (a) > R_t(b)$ and $\\\{ R_t (a) > R_t(b) \\\} =0$ otherwise.
>
> > Some of the plots are covered by the legends.
>
> We thank the Reviewer for pointing out the typo. We will change the dimension of the legend in the final version of the paper.

---

> > ### Comment · Reviewer_qbW7 · 2023-11-21
> > **Rebuttal Reply**
> >
> > I appreciate the detailed response from the authors, which addressed some of my concerns. However, I am still confused about some of the points I mentioned in the review.
> >
> > I might not express my thought in a proper way in the review, but in Theorem 1, do we have to worry about the impact of function $C(\cdot)$ in the lower bound? It is not about how the funciton is shaped, but how the function affects the bound. For example, is function $C$ dependent on time horizon? Can we obtain some bound on $C$? Where does it come from? In fact, we have zero information about $C$ except its domain. This could be improved if there is more discussion about the theorem. In my case, the proof sketch did not help on understanding how the bound is obtained.
> >
> > The explaination to Theorem 4 makes some sense to me, but what I am worried is whether the mentioned events in the rebuttal also stand in expectation and whether it causes a trade-off between instance-dependent/independent bounds. Anyways, the authors mentioned that understanding this theorem is crucial, then they should present some discussion in the paper, since the result is counterintuitive.
> >
> > Nevertheless, the response to the design of Algo 2 is very clear and helpful, and now I understand how the algorithm works. In conclusion, as also mentioned by Reviewer qw3s, I feel like a noticable amount of theoretical results in the paper is lack of intuition, clarity, and possibly soundness. Although I believe that the authors have put solid effort in this work and I really appreciate it, I cannot just let it slip away this time as I do not think it is ready to be published. I believe reorganizing the writing would make readers easier to understand and appreciate the work.

---

> > > ### Author Response · Authors · 2023-11-21
> > >
> > > > Theorem 1
> > >
> > > The function $C(\cdot)$ cannot depend on $T$. As it is easy to check by reading the theorem statement, stating $\forall C(\cdot) \ \ \exists T: \dots $ implies that the time horizon $T$ cannot depend on the function $C(\cdot)$. Furthermore, the question about the shape of the function $C(\cdot)$ is ill-posed, since it is similar to ask which is the value of $\epsilon$ in the definition of the limit of a sequence of real numbers. Understanding these arguments does not require a complete grasp of the paper's proof, but rather familiarity with standard mathematical notions, such as First-order logic.
> > >
> > >
> > >
> > > > Theorem 4
> > >
> > >  The idea of an event holding "in expectation" is meaningless. Indeed, an event holding with a probability which depends only on $\Delta$ (non-negligible) results a challenge for any algorithm. The proof involves identifying two $\Delta$ values where both instance-dependent and instance-independent regret bounds fail. We think this intuition is enough, if it was necessary to give the full intuition of every proof in the main paper, no theoretical paper could be published at ML conferences.

---

### Meta-Review · Area_Chair_hyyk · 2023-12-12

**Metareview:**

This paper looks at multi-armed bandit problems, where the feedback is not the usual one but only a ranking on the average rewards gathered by each arms.

There has been a large discussion between the reviewers and myself about the setting and the results.

We believe that the main results are correct and, at the end, that the feedback system makes some sense.

The major concerns that we had are that the paper is not easy to read, at all, and it seems that the instance-dependent bounds only hold for Gaussian rewards (as the proofs are based on Brownian motions). We believe that it might certainly be possible to approximate sub-Gaussian rewards (say with some concentration inequalities or CLT) with Gaussian ones, but it's not clear to us what would be the implications.
Also, the literature review (on bandits) is far from being complete and is actually incorrect. We urge the authors to credit the correct original authors to the different concepts used (and not just refer to the latest book).

This paper just need some polishing, but more than what would be acceptable for acceptance this year, which is why we recommend rejection at the moment.

**Justification For Why Not Higher Score:**

N/A

**Justification For Why Not Lower Score:**

N/A

---

### Decision · Program_Chairs · 2024-01-16

Reject